# Sharing Cities and Commoning: An Alternative Narrative for Just and Sustainable Cities

**Adrien Labaeye**

Geography Department, Faculty of Mathematics and Science II, Humboldt-Universität zu Berlin, Unter den Linden 6, 10099 Berlin, Germany; adrienlabaeye@gmail.com or labaeyea@hu-berlin.de

**Abstract:** Sharing Cities are emerging as an alternative narrative which promotes sharing as a transformative phenomenon for just and sustainable cities. This article shows that Sharing Cities are conceived within the alternative political economy of the commons. Bringing a theoretical contribution into dialogue with a practice-oriented book, this paper aims at checking the concept of Sharing Cities against the reality on the ground by reviewing 137 secondary cases: (1) Is communal (non-commercial) sharing a substantial phenomenon? (2) What is the role of technology—and more widely, of intermediation—in sharing practices? (3) If at all, what is being transformed by sharing practices? (4) Are commons depicted in each case? Results show that most cases display a communal form of sharing that is independent of digital platforms, i.e., that the sharing transformation affects all arenas of production and social reproduction across a wide variety of sectors, and it relies on translocal replication rather than up-scaling. With only 26% of cases apparently depicting a commons, this paper argues for a relational epistemology of urban commoning, shifting the focus to more-than-human commoning-communities. Thus, Sharing Cities are captured not only as a set of policy proposals and practices but as the performative depiction of an alternative worldview based on interdependence, ready for the Anthropocene.

**Keywords:** sharing cities; sustainability; commons; commoning; smart cities; digital technology; case study review; grassroots innovation; anthropocene

---

## 1. Introduction

The Sharing Cities approach is emerging as an alternative to the smart city discourse on the opportunities created by digital technology at the intersection of cyber space and urban space [1,2]. At a time when urban settlements are seen as the locus of the sustainability challenge [3–5] it has been argued that cities—through population density and a highly networked urban space—provide critical mass in both demand for and supply of shared resources and facilities [1]. With the rise of digitalization, so-called sharing economy online platforms that enable collaborative consumption and production are also reshaping urban economies [6]. Boyko et al. [7] argued that the sharing economy presents a very narrow view of the potential of sharing to contribute to more sustainable cities. In contrast, the Sharing Cities discourse includes, but also transcends, the sole sharing economy approach, acknowledging that in cities, both commercial and communal forms of sharing may coexist [1,7]. The Sharing Cities narrative has also been identified as an emerging counter-narrative to the neoliberal imagery of urban development [2]. With Sharing Cities, Duncan McLaren and Julian Agyeman [1] argued that a "sharing paradigm" rooted in the political economy of the commons—beyond market and state—can be a transformative force for more just and sustainable cities.

In 2018, the non-profit online media platform shareable.net—"A valuable ally in our evolving thinking" [1] (p. 18) and catalyzer of the Sharing Cities movement [8]—released its own Sharing Cities publication [9]: A practice-oriented book fleshing out the alternative narrative with a collection

of 137 short case studies depicting local bottom-up initiatives and policies. It adds to preliminary mappings of sharing in cities executed in specific locations [7].

Notably, both publications have anchored their narrative of Sharing Cities in the political economy of the commons. This concomitance provides a unique opportunity to map a broad scope of field practices in the light of the analytical frames proposed by McLaren and Agyeman [1] and get a clearer view of what this emerging Sharing Cities narrative is made of. Thus, in this paper, we review the case studies assembled by Shareable and its collaborators, using McLaren and Agyeman's approach [1].

## 1.1. Rooting the Sharing Cities Discourse in the Political Economy of the Commons

The sharing paradigm as proposed by McLaren and Agyeman [1] is based on an understanding of well-being that requires building and developing human capabilities for all. According to the authors, "the fundamental resources we have available to do that [ . . . ] are better conceived and understood as shared commons than as private goods" [1] (pp. 8–9). This centrality of the commons as a form of economic organization in the Sharing Cities narrative has also been identified elsewhere [2]. Consistently, Shareable's Sharing Cities ambition is to dissipate the "blindness" to people's power to meet their needs outside of the market and state through the commons [9]. Following Elinor Ostrom [10], it is less about seeing the commons as a new panacea for urban development and more about bringing back the three spheres of market, state, and the commons into balance and harmonization, with each one controlling the excesses of the others [9]. It answers a concern that much of the sharing economy literature frames sharing only as an economic activity rather than a social, cultural or political one that is rooted in urban environments [1,7,11]. To reflect this diversity, McLaren and Agyeman [1] propose that the sharing paradigm can be mapped on a continuum between commercial and communal sharing.

Under the commons paradigm, a sharing city is not only about the right for urban dwellers to use shared resources and infrastructure; it is, fundamentally, a right to remake cities [1,12]. Thus, in her seminal work, Elinor Ostrom outlined a series of eight conditions for a successful governance of the commons including the rights of individuals affected by a resource regime to participate in making and modifying its rules [13]. Another condition is that this right of the citizens to make rules directly is recognized by the government [13]; a condition that is generally not met in the urban context but which can change, as illustrated by the pioneering city government of Bologna in Italy, where a law was passed to create opportunities for citizens to be directly engaged in the management of urban commons [14].

The implementation of the sharing paradigm calls for socio-cultural and political changes [1]. Thus, by rooting the sharing paradigm in the urban space and its politics, the sharing cities narrative may avoid "the post-political trap" [1] of utopian discourses of smart cities where digital technology would fix major urban issues [15–17]. In this line, the sharing city approach may have strong commonalities with the municipalist movement in Spain that has often used the concept of commons as a central element of local political platforms, such as Barcelona's En Comú. Indeed, "the conceptual flexibility and diversity of understandings of the term [commons] offers the hegemonic potential of serving as a cornerstone for a political project that, on the one hand, rejects neoliberal privatization, and, on the other, refuses to fall back into the kind of monolithic understandings of the public/the state" [18] (p. 21).

## 1.2. The Problematic Role of (Digital) Technology

One of the stated aims of McLaren and Agyeman's *Sharing Cities* book is to show how "truly smart cities must also be sharing cities" [1] (p. 2). Indeed, the smart city discourse has been largely criticized by observers and academia for imposing a technocratic and market-driven vision of city governance on local governments and their citizens [16,17,19]. McLaren and Agyeman [1] (p. 5) want to redefine smart cities as a way of "harnessing smart technology for an agenda of sharing and solidarity, rather than one of competition, enclosure, and division." However, they do not restrict sharing to practices using (digital) technology and present in their sharing paradigm a continuum between inter-mediated

forms (through a third party such as using an app or similar) to informal or socio–cultural sharing (with no third party involved, such as sharing between friends or neighbors) [1]. This is a major distinction from the usual sharing economy literature that gives a defining role to digital technology as a key enabler for collaborative consumption and production [20–22].

Rather than with technology per se, they see the key issue lying with the distribution of power around the organizations using such technology [1] (p. 118). However, while power is certainly a core issue, it has been argued that (digital) technology is a problematic category in itself [15,17]. Thus, Kitchin [17] and Gitelman and Jackson [23] have shown that data—the core resource of mainstream smart cities—are never raw and never neutral. Morozov [15] argued that algorithms—the core technology to process data—are bound to present shortcomings or biases. It appears that the problem with technology does not only lie in the power structures surrounding its use but also in the wider political–economic context it is embedded into (March 2016). Thus, the smart cities narrative has also been characterized as an overly techno-optimist vision [15,24]. It is therefore necessary to problematize both the technologies that are heavily shaped by commercial or state actors and the potential contradictions that may appear in their application to enable commons-oriented initiatives and practices. Additionally worth considering is that sometimes a certain "penchant for technological solutionism," as put by Morozov [15] may drive observers to overestimate the role of technology above the importance of community-based actions in commoning processes and practices, even when they partly rely on the use of digital tools [25].

Spanish municipalist governments such as Madrid and Barcelona are at the political vanguard of the movement to reclaim the urban digital infrastructure. Thus, in Madrid, an open-source platform (decide.madrid.es), launched in 2014, enables citizens to submit and select projects to be funded under the participatory budgeting process [18]. As shown by Rubio-Peyo [18], through the generalization of open data and open source software for all city operations, the municipality of Barcelona has locally redefined the notion of "smart cities" away from what Kitchin pointed as "an underlying neoliberal ethos" [17]. This echoes efforts to redefine technology away from productivist capitalism: Thus, an "appropriate technology" is owned by the local community [26]. With the digital transformation, this has naturally found a declination as open source appropriate technology [27]. It is unclear what role such alternative technology has and under which concrete form it may play in Sharing Cities.

*1.3. The Transformative Power of Sharing*

Referring to Shareable's founder Neal Gorenflo, Ede [28] and McLaren and Agyeman [1] showed that beyond transactional sharing, which is of mostly economic nature and focused on improving efficiency of asset use and cost-sharing, transformational sharing involves a shift of power and social relations; it emphasizes solutions that build residents' ability to work together [9]. Thus, they argue that the intangible benefits of transformative sharing are of potentially greater significance than the tangible ones resulting from transactional sharing [1] (p. 255). For Sharp [8], the transformative sharing as promoted by Shareable since 2013 through the Sharing Cities network is based on community empowerment and grassroots mobilization, qualifying as a transformative social innovation [29]. The latter is defined as a "social innovation process that challenges, alters, or replaces existing institutions and institutional arrangements across the context (i.e., in more than just a single isolated social experiment)" [29] (p. 11). By many accounts, the sharing paradigm deployed in cities by McLaren and Agyeman [1] or Shareable proposes to transform institutions along the whole spectrum, from somewhat private interactions to municipal rules: "In conclusion therefore, as we understand it, sharing offers both a sustainable foundation for participatory urban democracy and a transformative approach to urban futures." [1] (p. 322)

To analyze the scope of such a transformation, McLaren and Agyeman [1] refer to Harvey's [30] seven arenas (norms, rules, values, etc.) in which neoliberal capitalism—or, alternatively, the sharing paradigm—shapes life: Forms of production, exchange and consumption; relations to nature; social relations between people; mental conceptions of the world, embracing cultural understandings and

beliefs; labor processes; institutional, legal, and governmental arrangements; and the conduct of daily life that underpins social reproduction [30].

A main rupture with the sharing economy approach and a core feature of the Sharing Cities discourse is to advocate for extending the sharing paradigm beyond mere bike-sharing schemes and other accommodation policies for Airbnb to the whole city as a system in all its dimensions, including financial, institutional and cultural ones: "We suggest that "sharing the whole city" should become the guiding purpose of the future city" [1] (p. 5). This idea echoes strongly the work of Foster and Iaione [31] who have suggested reconceiving "the city as a commons" by transforming the role of the local state from one of a regulator to one of a facilitator of citizens' direct involvement in the governance of shared assets and municipal services. They argue that the city is a commons by virtue of its openness, resulting in a potential for rivalry as well as producing collective wealth [31]. In that context, "sharing"—similarly to "commoning"—is understood as a third way of governance and provision, rooted in the collective governance of jointly held resources [1] (p. 14).

In their introduction McLaren and Agyeman [1] stated the ambition to show how a broad understanding and implementation of sharing can overcome the shortcomings of commercial approaches and transform our understanding of sharing and cities. While substantiated with empirical evidence, it is mainly a theoretical effort. In turn, the motivation of Shareable's *Sharing Cities* book is to bring the already existing pieces of the puzzle together so that the sharing city becomes a more concrete vision [9]. How far do those empirical elements substantiate or contradict the conceptual approach laid out by McLaren and Agyeman [1]?

## 2. Research Questions

To explore this main line of investigation, four questions are emerging.

McLaren and Agyeman [1] stressed the importance of communal (community-oriented) sharing as a transformative force. On that basis, how much communal, as opposed to commercial, sharing practice is there in the field? Given that Shareable's [9] collection of cases was also motivated by drawing attention to commons-oriented initiatives, it is reasonable to hypothesize that a large share of the cases to be reviewed will, indeed, depict communal sharing practices.

It has been shown previously that while sharing is often associated with the emergence of digital technologies, the sharing cities approach also includes practices that are not inter-mediated online. To gain a clear view of the role of digital intermediation in the Sharing Cities narrative, this article investigates how cases are distributed along the inter-mediated/socio-cultural continuum. From Shareable's focus on people as key actors, it can be expected that technology is playing a less important role in the sharing cities discourse than in the sharing economy, where the online platform is generally accepted as a defining feature [20,21].

Using the sharing spectrum laid out by McLaren and Agyeman [1], this paper will investigate what domains are impacted by sharing practices and policies: In other words, what is being shared? Is the scope of transformation as broad as it is suggested? The literature review showed that it may well be so, shifting our view of city governance beyond the sole consumption and production processes usually depicted in the sharing economy literature. However, the degree of commitment required at the political level of city governments to engage in integrated processes such as in Seoul, Bologna or Barcelona suggests that it is unlikely that many cases or policies will depict a cross-sector change in formal legal and governmental agreements. Therefore, it is expected that most cases have an impact—if at all—in narrow fields of policy and, possibly, on "soft" institutions such as norms and values.

McLaren and Agyeman [1] emphasized that key urban resources are better conceived of as commons than private goods. The transformation in the governance of urban resources towards community-managed commons is therefore critical to the Sharing Cities narrative. However, on the ground, are such urban commons already a reality? In the introduction to Shareable's book, Gorenflo warned that few cases are "purely commons oriented," some only have commons elements, and others just set the stage for commons development [9] (p. 29). To assess the empirical relevance of urban

commons as key building blocks of the Sharing Cities narrative, it appears critical to check whether an actual commons—a set of relations between a resource, a community, and rules—is being depicted in each case.

## 3. Methodology

In this paper, the 137 cases and policies compiled in Shareable [9] are reviewed using two main analytical representations proposed by McLaren and Agyeman [1] to navigate the field of sharing cities as well as insights from the commons literature to check for the presence of a commons in each case. Some degree of subjectivity is inevitable in the analysis of the limited material at hand while scoring and sorting the cases: Here, only the material provided in the book is taken into consideration to increase reproducibility of results. To limit subjectivity, research questions are operationalized through concrete closed questions, in particular in investigating the sharing paradigm. Though not purely positivist, I believe this review provides a useful basis to discuss the narrative of Sharing Cities and its rooting in existing practices within the limitations of the material provided by Shareable's book. The detailed results and scoring database are available upon request to the author.

### 3.1. Mapping the Sharing Paradigm

One analytical tool mobilized is a mapping of the sharing paradigm (Figure 1) along two axes into four "flavors of sharing." The horizontal axis represents the continuum extending from inter-mediated (i.e., through platforms or third parties) to socio–cultural sharing, and the contrasting poles of commercial and communal sharing are on another axis [1] (p. 13). The authors stress that this characterization is a gradual one and speak of four "flavors," graduations of sharing combined as in Figure 1.

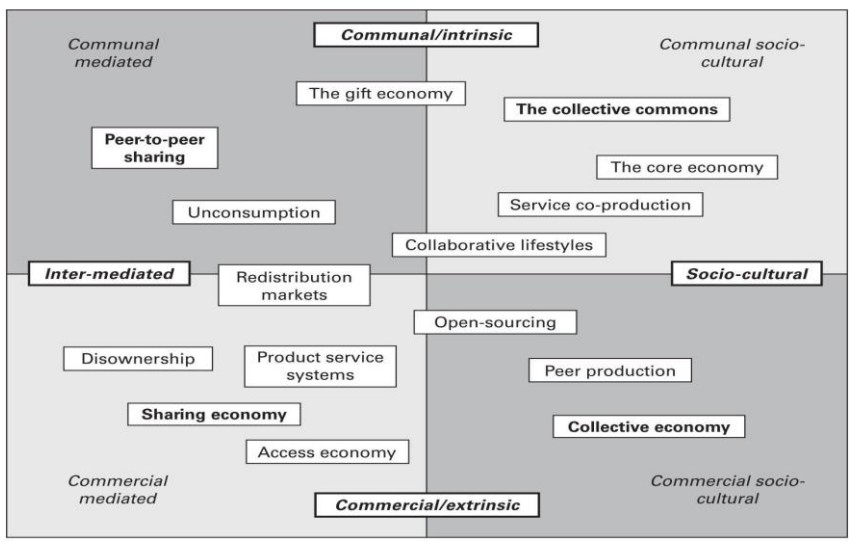

**Figure 1.** Key dimensions of sharing [1].

To reflect the gradual nature of this classification, the present review sorts the cases along two axes with a simple scoring. The horizontal axis contrasts inter-mediated (on the minus side) vs. socio–cultural (on the plus side) practices. The vertical axis sets commercial on the minus side and communal on the plus side. This coarse scoring intends to show some contrast in the degree of which initiatives display the various attributes ("flavors"). It is operationalized by asking, for each dimension (corresponding to an axis), four questions that build upon the developments of McLaren and Agyeman [1]. Each "yes" scores +1, each "no" scores -1, except for the first and main question for each dimension, where the score is doubled to ensure appropriate contrasting.

*"On one dimension, we see a contrast between sociocultural or informal sharing (typically between family members, friends or neighbors, directly organized by the participants in line with social norms) and (inter)mediated sharing, which is mediated through a third party (often using a website or mobile application)."* [1] (p. 14)

Thus, for the horizontal axis inter-mediated/socio–cultural, the questions are the following: (1—main question) is the sharing practice socio–cultural (i.e., happening without going through an external third party, requiring digital platform or not)? (2) Is the sharing practice possible without using a digital platform? (3) Are the shared resources co-owned by the participants themselves? (4) Is the sharing practice the result of a co-evolved tendency shared by a group (rather than a learned behavior/replicated initiative)?

*"The other dimension is about why we share and the motivations of the participants. On this second axis, we map a contrast between typically extrinsic motivations, notably commercial gain, and intrinsic motivations based in a sense of community, which we label as the commercial–communal axis."* [1] (p. 14)

For the vertical axis of commercial/communal, the questions include: (a—main question) Is the initiative/practice mostly intrinsically motivated, i.e., based in a sense of community rather than commercial gain? (b) Are profit-oriented/commercial activities completely excluded from the practice itself? (c) Is the practice free of monetary transactions? (d) Are the participants involved in some sort of self-governance (e.g., co-shaping the rule/norms applying to the practice)?

This analytical tool can only be used to review half the cases (69 out of 137), i.e., those portraying initiatives that depict a sharing practice and not public policies that present regulations, decisions, strategies, etc.

### 3.2. The Sharing Spectrum: What is Actually Being Shared?

Going one step further to substantiate their argument that sharing is transformative, McLaren and Agyeman [1] (p. 255) proposed a tool to map the sharing spectrum (Figure 2) according to the domains where sharing is deployed (i.e., what is shared), from more tangible domains to more intangible ones (Table 1). It connects these "sharing domains" to Harvey's [30] arenas of production and social reproduction where sharing may result in changing norms: "Forms of production, exchange and consumption; relations to nature; social relations between people; mental conceptions of the world, embracing cultural understandings and beliefs; labor processes; institutional, legal, and governmental arrangements; and the conduct of daily life that underpins social reproduction" [1] (p. 13).

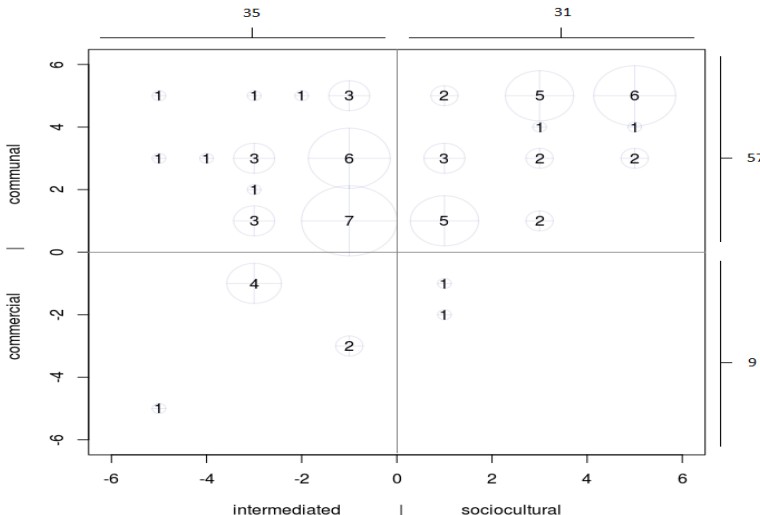

**Figure 2.** Graph representing the dispersion of the cases across the sharing paradigm (the number indicating the number of cases for each pair of coordinates; *n* = 66). Author.

**Table 1.** The sharing spectrum [1] (p. 254).

| Sharing Domain (What is Being Shared) | Concepts | Examples | Arena(s) Where This May Change Norms |
|---|---|---|---|
| Material Tangible | Industrial ecology | Circular economy, recovery and recycling, glass and paper banks and collection, scrapyards | Relations to nature; form of production, exchange, and consumption |
| Production Facility | Collaborative production | Fab-labs, community energy, job sharing, open souring, credit unions, and crowdfunding | Forms of production, exchange, and consumption; labor processes |
| Product | Redistribution markets | Flea markets, charity shops, freecycle, swapping, and gifting platforms | Forms of production, exchange, and consumption |
| Service | Product service systems | Ridesharing, media streaming, fashion and toy rental, libraries | Forms of production, exchanges, and consumption; labor processes; conduct of daily life; social relations between people |
| Experience | Collaborative lifestyles | Errand networks, peer to peer travel, couchsurfing, skillsharing | Conduct of daily life; conceptions of the world; social relations between people |
| Capability Intangible | Collective commons | The internet, safe streets, participative politics, SOLEs, citizens' incomes | Conceptions of the world; social relations between people; institutional, legal and governmental arrangements |

It is deemed that the self-explanatory nature—in particular thanks to the examples—of this analytical tool does not require any further operationalization step and can be used "as is" to review cases and policies. Thus, each case material was examined to determine what is being shared (the sharing domain—here only one option per case is selected) and what arenas it may impact (here multiple arenas are accepted for each case).

*3.3. Identifying a Commons*

In addition to the domains impacted by sharing, it is of interest to explore whether general sharing actually translates into the community-governance of resources, i.e., commons. Thus, as Gorenflo mentioned in the introduction of Shareable's book, not all cases presented are depicting a clear-cut commoning practice:

*"For instance, there are few cases and policies that are purely commons-oriented. The majority of the pieces have a commons element, and the rest arguably set the stage for commons development. For instance, Barcelona's Solar Thermal Ordinance (Chapter 5) helps to localize renewal energy production, setting the stage for a commons approach to energy, but doesn't imagine a commons in its effort to promote sustainability."* [9] (p. 29)

This ought to be reviewed in more detail in order to obtain a more precise notion of such proportions, with the aim of informing observers of the actual role of commoning in the Sharing City approach. Three options are foreseen contrasting the various importance of the commons in each cases. First, taking inspiration from Ostrom's definition [32], so-called "purely commons-oriented" cases are defined as those where (1) a clearly identified resource is being shared by a group of people who (2) manages it collectively (3) through a set of rules shaped beyond state governance or market mechanisms. Second, as shown in the literature review, commons are also increasingly being described as the relational social framework formed by the resource, the community and the rules [33]. Therefore, when either a shared resource, a collaborative practice, or a community is present (i.e., not at the same time), cases are ranked as (only) "having a commons element." Third, all other cases are categorized by default as "setting the stage for communing," following Shareable's authors, although this might be a matter of contention.

## 4. Results

### 4.1. Some Geographical Considerations

The book features cases from every continent, although none from south-east Asia, China, the Middle-East, North Africa, or Russia. In many of those places, state control over the economy and the civic society remains tight. Some cities often come back into focus. Thus, some of the cities that are often described as hot places of the sharing movement include San Francisco (five cases), Seoul (four), Barcelona (four), and, perhaps more surprisingly, London (five) and New York (four) that are usually more known as financial centers. Portland (four), Hamburg (three), and Helsinki (three) are also noticeable. As it could be expected given a largely Western authoring team, out of 137 cases, 44 are located in Western Europe and 42 are in the US and Canada. Though US–Canada and Western Europe are dominant, it is worth appreciating that they both amount to 62% of the cases, giving ample representation to other areas of the world such as South Korea, India, South Africa, Brazil and Australia. Nevertheless, while the sharing movement also spread in recent years to south-east Asia and China, these regions are missing from a book that could have benefited from a more comprehensive assessment. Yet, it is fair to say that Sharing Cities are a global phenomenon

### 4.2. Mapping along the Four "Flavors" of the Sharing Paradigm

The "four flavors of sharing" explore the dimensions of intermediation and motivation that drive the practice. Thus, out of the 137 cases contained in the book, the 67 policies do not qualify for assessment here because they do not depict a sharing practice. Though they were contained in the initiatives' section of the book, three additional cases were excluded from the analysis because they do not qualify as a sharing practice where one could answer the rating questions; these are the Stop Wasting Food Campaign [9] (p. 172), the Pittsburgh Community Bill of Rights banning fracking [9] (p. 186), and Bologna's Regulation on the Care and Regeneration of the Urban Commons [9] (p. 252). Here, the sample size is therefore reduced to 66 cases.

As seen in Figure 2, sharing cities as they are described by Shareable cover a set of practices that is largely communal (86% of the cases that are not policies). Nevertheless, the degree of communality varies, and cases are spread rather evenly along the communal dimension. Cases with a score of 3 usually answered negatively to question (c) ("Is the practice free of monetary transactions?") or (d) ("Are the participants involved in some sort of self-governance"?), and all but two completely excluded profit-oriented activities. Interestingly, five out of the nine cases that scored negatively on the vertical axis, and therefore had a rather commercial flavor, were actually rated as intrinsically motivated cases (in reference to question (a)). This leaves very few cases (four) where the main motivation was commercial rather than being rooted in a sense of community. Three of those are co-operatives, where ownership is shared between workers or investors. Consistently, with its introductory statement that "the commons needs to be elevated to a dramatically higher level of importance in urban development" [9] (p. 32), the selection of cases by the authors of the book has clearly favored a communal orientation for fleshing out what a sharing city can be. To nuance this assessment from a methodological level, it is useful to note that seven cases reviewed scored low ($y = 1$) on the communal–sociocultural quadrant ($x > 0$; $y > 0$). Examples range from collectives of social or cultural entrepreneurs to FabLabs and a wind-energy cooperative, examples where a soft commercial orientation is combined with a clear community purpose. These examples do give a taste of what socio–cultural and commercial sharing look like, although they are situated in the communal half of the quadrant.

Over half the (eligible) cases (35) display some degree of third-party intermediation, even in initiatives with a communal flavor. However, only a third of them use some sort of digital platform (23 cases out of 66). This tends to depict a different reality to a sharing economy that solely focuses on digital technology as a key enabler. In addition, when there is intermediation, it is low. From the material analyzed, intermediation can generally be found in two types of cases. On the one hand,

a third party may be necessary to organize a relatively complex service or to provide a significant infrastructure. This applies in the case of Humanitas [9] (p. 40), a senior housing project that also offers housing to students in exchange for volunteering with older people, or Regionalwert AG [9] (p. 234), a cooperative investment fund for sustainable local food. Here, an organization takes over the coordination of a complex new service (Regionalwert AG) or integrates sharing into an existing service (Humanitas). On the other hand, a third-party provider may be an instrument to scale up a practice. Thus, Freifunk [9] (p. 208) offers the platform and open source resources across Germany for local groups to develop their own alternative internet infrastructure. Using an online platform, CoAbode [9] (p. 46) matches single mothers for sharing flats and help with daily life across the US. Interestingly, when over half of the cases (34) are the result of coevolution by a (local) group, i.e., not a scaled-up practice—21 of these practices are still happening through a third party. In fact, 70% (46) of the cases involve a third-party. Therefore, sharing practices that are more sociocultural are a minority. Among various other reasons, an intermediary organization—an association or a cooperative—is often required for legal aspects, such as co-owning assets or managing liabilities.

Three clusters can be described. One covers cases that have a high degree of communality and are highly sociocultural. Such cases are Walking School Buses (low tech mobility solution), 596 Acres (reclaiming of vacant land for community purpose), Repair Cafés (repair self-help communities), Tarun Barat Sangh (community-led water management), Resident Development Committees (community-led water management), Depave (community-driven removal of impervious pavement), Incredible Edible (guerilla urban gardening), Les Murs à Pêches (cultural event for community activation), Bottom Road Sanctuary (self-governed nature reserve), Chisinau Civic Center (reclaimed land for community purpose), Water Management Beyond Politics (historical governance structure), Neighborhood Partnership Network (residents empowerment in planning), Club of Gdansk (multi-stakeholder forum for planning). Notably, four of these are involved with sharing that pertains to land, and four others are related to water. Another cluster appears around the center of the graph with a light intermediation flavor and rather communal orientation. Examples are: Humanitas (students co-living with seniors), Opportunity Village Eugene (transitional housing for homeless people), Miethäuser Syndikat (co-housing supporting organization), Seva Café (pay-it forward restaurant), Evergreen, Library at the Dock (maker space in public library), Platform Co-op (cooperative model for sharing platforms), Community purchasing alliance (pooling purchasing power), Human Ecosystem Project (reclaiming social media data), ShareHub (online information platform about sharing), RegionalWert AG (cooperative investment fund for local food), Nippon Active Life Club (time banking for seniors), Liquid Feedback (digital decision-making tool). A third, smaller cluster, in the lower left quadrant, can be described as having medium intermediation and a low commercial flavor. Three of these are about mobility: SafeMotos (addressing safety in mototaxi business), Multimodal Toolkit (encouraging multimodal mobility), and RideAustin (not-for-profit ride-hiring app), and one is about waste: Warp It Reuse Network (interorganizational marketplace for surplus office furniture and supplies).

With only two cases—COOP Taxi in Seoul [9] (p. 64) and Compost Pedallers in Austin, Texas [9] (p. 166)—sharing that is both socio–cultural and commercial (open sourcing, peer-to-peer economy) barely appears in the book. Again, the fact that the book largely focused on commons-oriented approaches may explain this. However, the absence of a now widespread practice such as coworking raises questions on how much existing and substantial evidence of this sharing flavor is missing from the picture of a Sharing City drawn by Shareable in its book.

### 4.3. What is Being Shared? Review of Sharing Domains along the Sharing Spectrum

More than two thirds of the cases involve either sharing of a production facility (36%) or of a capability (33%), see Table 2 and Supplementary. This dominance is striking. On the more tangible end of the sharing spectrum, a significant amount of practices involves production facilities. In some cases, though—such as housing—production was understood as re-production. Indeed, housing can hardly

be understood as a product to be consumed: It is a space, a facility, that serves the reproduction of social life, productive capacities, etc. Thus, often the (re)production facility is a shared space such as land or a building. However, as illustrated with the finance and work cases, facilities are not necessarily as tangible as a production site; they often are organizations. At the other end of the spectrum, 45 cases involve sharing in the capability domain, and 27 out of them are filed in the book as policies. Even out of the 16 cases that are ranked as initiatives, a good third of them have many qualities of a policy, e.g., the description of Bologna's law on the Care and Regeneration of the Urban Commons [9] (p. 252), the Club of Gdansk—a Cross-Sector Collaboration for Urban Administration and Planning [9] (p. 260), Pittsburgh's Community Bill of Rights Banning Fracking [9] (p. 186), and Water Management Beyond Politics in the Netherlands [9] (p. 190). For many policies, it was difficult to identify a sharing practice involving a tangible domain. Conversely, it was difficult not to consider that these policies often contributed to something more intangible: The equal capabilities of citizens to achieve a potential towards social justice [1] (p. 205).

**Table 2.** Sharing domains—sample size = 137. Some cases involve more than one domain.

| Sharing Domain Book Chapter | Material | Production Facility | Product | Service | Experience | Capability |
|---|---|---|---|---|---|---|
| Housing | | 7 | | 1 | 4 | 1 |
| Mobility | | 1 | | 9 | 1 | 2 |
| Food | | 2 | 8 | | 2 | 4 |
| Work | | 10 | 2 | 3 | | 7 |
| Energy | 2 | 7 | | | | 3 |
| Land | | 9 | | | 1 | 2 |
| Waste | 2 | 3 | 3 | | | 4 |
| Water | | 2 | | | | 10 |
| Technology | 5 | 1 | | 4 | | |
| Finance | | 11 | | | | 1 |
| Governance | | 1 | | | | 12 |
| TOTAL | 13 (9%) | 50 (36%) | 14 (10%) | 16 (12%) | 9 (7%) | 45 (33%) |

In water cases, capability is the main sharing domain (9). They depict initiatives and policies that empower communities to govern their water resources more directly and sustainably, whether through traditional institutions, such as in the Netherlands [9] (p. 190), in Pittsburgh, where the municipality banned fracking [9] (p. 186), or, for many cases located in the Global South, by building the capabilities of communities to self-govern the resource as shown by the NGO Tarun Bharat Sangh in Rajahstan, India [9] (p. 192). Similarly, governance stories report on tools and policies that empower communities to make decisions for themselves with online platforms such as the open-source app Loomio [9] (p. 258) used for collective decision-making by an artist collective, or the collaborative mapping of SynAthina launched by the municipality to network civic actors in Athens [9] (p. 254). Offline, neighborhood assemblies, for example, have also shown themselves as capable of integrating citizens in budgeting processes as in Porto Alegre, Brazil [9] (p. 262) or in city planning as in New Orleans after hurricane Katrina [9] (p. 259).

In the mobility sector, what is shared is mostly a service (9). From private initiatives like Ride Austin [9] (p. 70) that offer traditional ride-hailing, to municipal bike-sharing [9] (p. 76), car-sharing [9] (p. 77) or even comprehensive shared mobility strategies such as in Milan [9] (p. 78), common sharing services are offered, and sometimes bundled, often in a non-commercial way. The most dissonant case—and therefore radically innovative—is probably one that addresses mobility from the experience side with the Walking School Buses [9] (p. 66), showing that a sharing city may emerge solely from activated communities with zero infrastructure or tangible capital involved.

Food cases mostly involve sharing a product (8). Thus, in Dublin, FoodCloud matches stores with surplus food with actors like food banks that redistribute it to people in need [9] (p. 82). The Seva Café in Ahmedabad successfully serves meals for free encouraging customers to pay for the next client [9]

(p. 84). Sometimes, the product shared is not food but cooking tools as with Kitchen Share [9] (p. 90) or Seeds [9] (p. 93). In some cases (2), mostly with policies, the focus is on sharing production facilities. San Francisco's Urban Agriculture Incentive Zone enables citizens to get a tax reduction for growing food in their gardens in exchange for opening the production sites to the public (classes, customers, community gardeners) [9] (p. 92). The small French municipality of Los-en-Gohelle offered free access to public land to farmers on the condition that they grow organic food supplying the schools [9] (p. 96). Two policies are remarkable, as they support an intangible commons—a capability. Firstly, food sovereignty with nationwide agro-ecological strategy in the Cuba Agro–Ecological Strategy to Increase Food Sovereignty [9] (p. 94) and, secondly, local food security with urban family gardens for low-income households in Medellin, Colombia [9] (p. 100).

The sectors of Land (9), Finance (11), and Work (10) mostly involve sharing production facilities. Land cases often involved the re-appropriation of public assets by citizens to turn them into community managed space, such as a vacant land in New York with 596 Acres [9] (p. 148), in Moldova with the Chisinau Civic Center [9] (p. 154), or with Hamburg's Gängeviertel, where citizens reclaimed a historical yet depreciated neighborhood for art and culture purposes [9] (p. 153). Community land trusts in the UK [9] (p. 146) or in the US [9] (p. 156) enable low-income households to access property while keeping land itself outside of the real estate market to avoid speculation. What is shared may also be less tangible, such as in the City Repair case in Portland, Oregon, where inhabitants increased their shared capability over their neighborhood by collectively mobilizing and taking unilateral action to make a dangerous intersection safer and spreading the method over the city [9] (p. 160). In the finance sector, most cases involved a shared production facility that takes various forms: A credit union for underserved communities in Durham, North Carolina [9] (p. 237), a cooperative investment fund for local sustainable food production in Freiburg, Germany [9] (p. 234), a time bank for the elderly in Japan with the Nippon Active Life Club [9] (p. 236), and community currencies such as Conjunto Palmeira in Brazil [9] (p. 232), in Brixton in the UK [9] (p. 242), or in Switzerland [9] (p. 238). In the work realm, such facilities are often physical spaces: For digital fabrication with Fab Labs [9] (p. 106) or making in a public library in Melbourne [9] (p. 111). Shared facilities may take the form of a network of social entrepreneurs such as Enspiral [9] (p. 104) or an online platform owned as a cooperative [9] (p. 112). However, in many work-related cases (7) the sharing practice may also encompass building up capability. Thus, the One-stop-shop of Social Clauses in Rennes, France, is a municipal procurement contractual practice that requires social inclusion among its suppliers, thereby strengthening the capability of workers with low skill levels [9] (p. 120).

A handful of other sectors (energy, waste, technology) gather cases deploying a variety of sharing domains. The energy sector displays a combination of cases sharing a production facility (7), such as production cooperatives [9] (pp. 128, 132); sometimes a material (2), such as bundled energy purchases [9] (p. 130); or a capability (3) with feed-in tariffs encouraging renewable energy production or an oversight public trust ensuring profit sharing among all stakeholders [9] (p. 134). Strikingly enough, the waste chapter features only two cases where sharing involves the material, waste: The Compost Pedallers in Austin, Texas collect food waste with cargo-bikes to transform it into a valuable product—compost [9] (p. 166); and in Curitiba, the municipality gives out food or bus tickets for collected and separated waste [9] (p. 168). Others build capabilities (like the Repair Cafés) by helping people to repair their broken devices [9] (p. 170) or by supporting communities in self-organizing waste reduction and recycling, as in Johannesburg [9] (p. 176). A product might also be shared: A party-pack with reusable dishes to reduce event waste in Palo Alto [9] (p. 181), or an open-source tool to monitor waste production and benchmark solutions in Finnish municipalities [9] (p. 180). Interestingly, it is in the technology chapter that most practices where it is a material that is being shared can be found, and this material is data which are crowdsourced [9] (p. 213), reclaimed from social media [9] (p. 212), released as open data through municipal repositories [9] (p. 220), or licensed as Creative Commons on municipal websites [9] (p. 221). In this chapter, four of the cases

depict the sharing of a service, which, for two of them, is the alternative provision of internet access [9] (pp. 208, 216).

Eventually, in the housing sector, in seven cases, it is a re-production facility that is shared. Indeed, in collectively owning a house in Paris, as with the Babayagas, a self-managed co-housing for seniors [9] (p. 48) or the Mietshäuser Syndikat that enables co-housing across Germany [9] (p. 44), a very tangible asset is shared—a building—that is central in the social reproduction of productive capacities but most essentially of life in its broadest sense. Alternatively, housing may be provided as a shared experience as at Humanitas where students mingle with seniors [9] (p. 40), or at Opportunity Village Eugene, where homeless citizens can benefit from one of 30 transitional microhouses [9] (p. 42).

As noted in the introduction of this article, a central thesis of the Sharing Cities narrative is to understand sharing in its transformative dimension, beyond its usual framing as transaction [1]. For McLaren and Agyeman [1], the shift in power and social relations as well as an increase in value for all residents is what constitute the transformational nature of sharing. For Neal Gorenflo from Shareable, it lies in "solutions that build residents' ability to work together" [9] (p. 33). In this regard, with two thirds of cases that either involve the sharing of production facilities or capabilities—rather than just products or services—and most cases displaying some commons elements, this transformational quality is, at least partly substantiated.

### 4.4. The Arenas of Impact

Table 3 shows the distribution of arenas of production and social reproduction that are impacted by the case. Almost systematically, cases were estimated as having a potential impact on several arenas simultaneously. While the review attempted to restrict the analysis to the most direct and evident impacts, it is arguable for many cases that they have an impact on many of those arenas at the same time, if not all, as in the case of the Urban Agriculture Incentive Zone in San Francisco [9] (p. 92). This regulation provides fiscal incentives for citizens to use their land for agricultural purposes. It is a legal arrangement that intends to affect the structure and location of food production. Doing it may very well affect the cultural understanding of what a garden is, shifting from an aesthetic yet unproductive space to a source of sustenance. The conduct of participants' daily life is affected, possibly blurring the frontier between recreational and productive times. Producing food in a garden arguably transforms people's relation to nature, bringing attention to the environment as a living system, even more so that the plan in question also limits the use of pesticides and fertilizers. Allowing people to use their garden for professional agriculture is also a breach in the traditional urban zoning that separates activities, possibly impacting labor processes by facilitating hybrid professional activities by reducing the barrier of capital (cultivable land) to enter a new occupation. Eventually, the bill in San Francisco requires participants to open the land to the public and therefore will very likely affect social relations between people.

**Table 3.** Harvey's arenas of potential impact—sample size = 137. Many cases impact more than one arena.

| Arenas | Count | % (*n* = 137) |
|---|---|---|
| 1. "Forms of production, exchange and consumption" | 103 | 75% |
| 2. "Institutional, legal and governmental arrangements" | 89 | 65% |
| 3. "The conduct of daily life that underpins social reproduction" | 78 | 57% |
| 4. "Mental conceptions of the world, embracing cultural understandings and beliefs" | 69 | 50% |
| 5. "Social relations between people" | 53 | 39% |
| 6. "Relations to nature" | 52 | 38% |
| 7. "Labor processes" | 36 | 26% |

It can legitimately be argued that further operationalization in checking which arenas are affected would provide a more rigorous review and potentially more clear-cut results. Nevertheless, this exploratory review clearly shows that sharing approaches tend to be transversal, not limited to one arena of production and reproduction, and that no arena is left untouched. Noticeably, many of

the cases potentially have an impact on institutional, legal and governmental arrangements beyond the cases that are policy descriptions. It is also worth mentioning that labor processes seem to be the least affected (only in 26% of the cases) by the sharing practices and policies described; an observation that is somehow consistent with the fact that very few cases operate on a commercial basis. This observation is an important limit to the transformative reach of sharing, labor processes being at the center of monetary economic activity. Additionally, with only 38% of the cases addressing relations to nature, the Sharing Cities approach seems not to place a systematic focus on ecological sustainability in frontal contradiction with premises as defined by McLaren and Agyeman [1].

### 4.5. Is There a Commons, Really?

There are 35 cases (26%) where a commons could be clearly identified. Present in seven case studies, the land chapter is the one displaying the most examples that are "purely commons-oriented." Whether it is a nature sanctuary recovered from degradation by its neighbors in Cape Town, South Africa [9] (p. 152), housing land owned by a Community Land Trust in London [9] (p. 146) and Burlington, Vermont [9] (p. 156), or the design of an intersection made safer by its inhabitants in Portland, Oregon [9] (p. 160), a community of urban dwellers has direct and collective agency on a specific urban resource that plays an important role in the their daily lives. The physicality of land may play a role in facilitating the re-appropriation of resources. Though, in the mobility chapter, the only one case that really qualifies as commons-oriented is intangible: The commons is a "walking school bus" organized by parents themselves [9] (p. 66). The second chapter, by its count of pure commons cases, is the technology one. There, new resources have usually been created by a community: An infrastructure supporting the world's largest mesh network in Germany [9] (p. 208), crowdsourced environmental data supported by an open-source sensor, the Smart Citizen Toolkit [9] (p. 213), and a community coding commons with the Bloomington Coding School in Indiana [9] (p. 206). Three cases also depict how communities have created a finance commons through a local currency supporting their local economy [9] (pp. 238, 242, 232). In all these cases, very few (four) depict a long-standing (i.e., over two decades) commons: One of these is Begum Bazaar, a high-street where merchants have probably been self-regulating for centuries and, lately, endured the extreme and pro-car urbanization of Hyderabad, India [9] (p. 162). Expectedly, most such "pure commons" are depicted in the book as initiatives and not policies. Though, very seldom, some cases ranked as policies are actually community-run and completely qualify as a commons-oriented initiative: The particularity is that they are recognized and supported by the local government. This is the case of the Brixton Pound, a local currency scheme started at grassroots level, which now has its own mobile electronic payment system and is recognized for paying local taxes [9] (p. 242). In Paraguay, community-based sanitation boards that are fully endorsed by the state enable residents to directly self-manage water and sanitation services [9] (p. 196).

Sixty-three cases (46%) contain some commons elements. Many of these have strong commons features, the strong role played by market mechanisms or state institutions exclude qualifying them of pure commons. On the one hand, cases describing a cooperative are generally classified here, as they are largely market actors even though they have many qualities of a commons. One good example is the Community Solar Gardens in the State of Minnesota, where the state facilitates the process of acquiring shares in a solar energy cooperative for people who do not own assets where solar production capacity could be installed [9] (p. 140): There is a shared ownership of a resource but through a cooperative, a market mechanism. On the other hand, in cases such as municipal open-source software in Munich [9] (p. 218) and Grenoble [9] (p. 222), open data in Montevideo [9] (p. 220) and Rotterdam [9] (p. 159), in spite of the existence of a clearly defined shared resource, the fact that the local government is the main actor governing and maintaining the resource was thought to exclude them from "purely commons-oriented" cases. In plenty of other cases, the existence of a third-party provider is often justifying why an initiative is thought of as having only "elements of a commons." Thus, the CoAbode platform matches single mothers who look for a flat share to facilitate mutual support [9] (p. 46). The flat share that results from the matching is a commoning practice. However, the case focuses

less on co-housing space than on the online platform, the management of which is taken care of by a non-profit, not the users themselves. Examples of such intermediation abound: In Kigali, Rwanda, SafeMotos has built and manages an online platform to rate motorbike taxi drivers with the aim of strengthening the commons of safety in that specific business, Embassy Network offers flat shares for purpose-driven young professionals around the world [9] (p. 49), and Opportunity Village Eugene provides transitional housing for homeless people [9] (p. 42).

Thirty-nine cases (28%) describe an initiative or policy that set favorable conditions for the emergence of commons-oriented approaches/practices. Policies that directly support the emergence of commons-oriented models, such as cooperatives in New York with the Worker Cooperative Business Development Initiative [9] (p. 115), or all-encompassing policies such as in Seoul [9] (p. 114), Barcelona [9] (p. 116), or Bologna [9] (p. 252), which promote unambiguously and specifically sharing and commoning practices in many sectors and dimensions. These may also be policies that do not conceptualize a commons but are putting in place a framework that may be favorable to commons-oriented practices and organizations such as in the UK with the Public Services (Social Value) Act 2012 [9] (p. 118), various participatory local policy practices [9] (pp. 266, 52), or, as mentioned by Gorenflo in the book's introduction [9] (p. 29), policies that may create the conditions of a commoning practice by re-localizing the (re)production of a resource like food [9] (pp. 94, 92), energy [9] (pp. 139, 136), or, in the case of water, the re-localization of resource ownership through privatization reversal, as in Paris [9] (p. 202) and Bolivia [9] (p. 198). In the finance sector, credit unions [9] (p. 237) or banking services with a community- or public-purpose [9] (pp. 247, 240) may provide favorable conditions towards strengthening the intangible commons of accessible financial services.

Results have shown that 70% of cases do include at least some commons element, and 26% have a clearly-identified commons. Still, the book is a clear contribution to address what Gorenflo in the introduction called people's blindness to the commons option [9] (p. 27). With a vast majority of communal cases, it is also a unique, substantial, and empirical contribution to McLaren and Agyeman's argument that the sharing paradigm is not only an economic activity but also a political and cultural one [1] (p. 9). By displaying communal solution-oriented cases in sectoral chapters such as water, energy, food, work, or housing, Shareable strongly echoes the idea that instead of automatically turning to markets or states to solve "problems," we could look at our primary needs in cities and "the whole range of ways in which we can enhance human well-being in just and sustainable ways" [1] (p. 9).

## 5. Discussion

This section discusses three key issues of the Sharing Cities narrative in light of the existing literature: The role of digital platforms; the transformative nature of sharing; and the epistemological foundations of commoning as encompassing paradigm.

### 5.1. The Role of Digital Platforms in Sharing

The initial framing of Sharing Cities at the intersection of the cyber space and urban space [1] (p. 1) tends to suggest that digital technologies would play a central role. However, our results show that only a third of the cases assembled by Shareable are digitally-based, further making the case that the sharing is not limited to digital platforms [7]. This also situates the Sharing Cities discourse out of reach of the technological solutionism critique articulated by Morozov [15] or Kitchin [17]. Furthermore, cases where digital technology was involved have often featured open source software stacks, giving ownership and agency back to communities. This encourages bridges with scholarship that has conceptualized the role of open source technology to play a key role in sustainable development: See open source appropriate technology [27] and cosmo-localization [34].

Within the sharing movement, the question of the ownership of digital technology has led observers to describe the sharing economy as a "neoliberal nightmare" [35] or "neoliberalism on steroids" [36]. In response, the search for alternative models is mostly discussed around the Platform Cooperativism concept, putting the question of platform (cooperative) ownership at the center [9] (p. 112); [37].

Co-opted by large capital, sharing platforms are said to have aligned to mainstream economic imperatives (growth, consumerism and profit maximization) obliterating their initial promise for equity and sustainability [36]. Responding to the fact that cities are on the frontline in dealing with the disruption of the sharing economy, Scholz [37] and Schneider [38] argued that platform cooperatives could show as particularly relevant for municipalities: These and their communities could globally pool resources to create shared software platforms and locally manage sharing businesses such as short-term rentals to keep the value generated in local hands.

However, these community-owned platform cooperatives are mostly in the project stage and still need to prove they are actually working beyond isolated experimental ventures. Platform Cooperativism has also been exposed to two further lines of criticism. On the one hand, proponents of Open Cooperativism [39,40] insist on maintaining technical infrastructure as an open commons as a safeguard, arguing that the cooperative model has not prevented many organizations from mimicking global corporations in their market behaviors, organizational cultures, and management styles. On the other hand, platform cooperatives are still a third party—an intermediary organization. In contrast, the emergence of the blockchain technology opens up the possibility for commons-based peer production to emancipate from platform third parties [41]. Thus, for many observers of the sharing economy, the distributed blockchain technology and the Internet of Things will enable the disruption of big centralized platforms and truly unleash the potential of peer-to-peer economic transactions [42–44]. Nevertheless, critics have stressed that traditional issues of power and collective ownership cannot be "programmed away" [45]. This tends to be confirmed by the recent story of Arcade City, the foremost example of a city-oriented blockchain application that had positioned itself as a distributed alternative to Uber and was faced with major issues of ownership which derailed the initiative [46]. In this light, the editor's decision to leave out blockchain and the Internet of Things from Shareable' book to instead favor approaches like Platform or Open Cooperativism which are focused on sharing ownership rather technological innovation seems savvy and should inspire further research.

This is particularly important when the performative nature of discursive resources that describes new experiments is taken into account [47]. Not ceding to the sirens of technological solutionism on a sharing scene saturated with tech hype appears as a "discourse of economic difference" as put by Gibson-Graham [47] or Healy [48] in conceptualizing the search for alternative or diverse economies. Thus, while a certain number of scholars are busy developing an enlightening critique of the smart cities discourse [17,19,49–51] or rethinking it [52], the endeavor to put forward a truly alternative narrative of how digital collaboration may contribute to just and sustainable cities—e.g., Sharing Cities—could take inspiration from the rather low-tech approach found in Shareable's [9] effort and alternative models such as Platform or Open Cooperativism.

## 5.2. The Transformative Potential of Sharing Cities

McLaren and Agyeman [1] and Shareable's [9] understanding of the transformative nature of sharing lies on shifting power relations in favor of communities. However, existing literature addressing the transformational nature of social innovation has also stressed the criteria of translocality as determinant [53]. Thus, transformative social innovation (TSI) is defined as "a social innovation process that challenges, alters, or replaces existing institutions and institutional arrangements across the context (i.e., in more than just a single isolated social experiment)" [29] (p. 11). Some of the cases such as Repair Cafés, FabLab, or Walking School Buses that are presented as translocal cases [9] ought to be qualified as TSIs. However, many other cases—approximately half—have locally co-evolved. In contrast to the sharing economy and the global up-scaling of its platforms powered by billions of dollars in capital, various authors have noted that social innovations are rather prone to be replicated [54] or scaled out [55], multiplying and adapting the same ideas and process across locations, enabling them to stay true to their original values [56]. Analyzing the Sharing Cities movement, Sharp [8] observed that Shareable has catalyzed grassroots actors in replicating successful experiments. Niche resources and intermediary organizations are indeed known to play an important role in the

diffusion of grassroots innovations [56,57]. In this context, *Sharing Cities* [9], with its cookbook style, clearly adds to niche resources known to play an important role in the diffusion of grassroots. It also offers a generative and practice-oriented narrative of change [29], and, as a discourse of economic difference, it can be interpreted as a performative ontological intervention [47].

Bringing the narrative to an institutional level, McLaren and Agyeman [1] were also suggesting to "share the whole city" [1] (p. 5) by referring in particular to Seoul's far-reaching pro-sharing policy. Similarly, [31], building upon their catalyzing work with the city of Bologna [14] and its Regulation between Citizens and the City for the Care and Regeneration of Urban Commons, have argued for thinking of the city itself as a commons: An institution for collective action. These two instances of a city that is scaling commoning to a strategic level are duly reported in Sharing Cities (2018). Only one other such case is made mention of: The Barcelona ProCommuns policy, initiated by the municipalist coalition Barcelona en Comú led by housing activist Ada Colau. Yet, in these accounts, little attention is given as to how to build the local political leadership required for raising commoning to such a level on the city agenda. Those three cases displayed a singular political dynamic: In Seoul, the mayor was a long-standing civic rights advocate; in Bologna, the whole region of Emilia-Romagna is known for a long tradition of public support of the cooperative sector [58]; and, in Barcelona, as Rubio-Peyo [18] reports in an analysis of municipalism in Spain, the local council was elected as part of a country-wide movement of "political confluences" bringing together the commons approach and Bookchin's libertarian municipalism [59]. Thus, referring to such cases the Sharing Cities discourse positions itself beyond the "post-political trap" of economic or technological determinism [1,17] but somehow fails to provide a reproducible approach. For Bauwens and Niaros [60], who have identified similar commons-oriented political coalitions in the cities of Frome, Milan, and Ghent, the horizontal and translocal dynamic of bottom-up commoning initiatives needs to be completed by a vertical political dynamic that remains participative [60]. On this somewhat blindspot, the urban commons literature could learn from the transition management literature applied to urban contexts that has explored ways to facilitate the local upscaling of transitions initiatives [61].

### 5.3. Towards Commoning as a More-Than-Human Politics for Sharing Cities

With 70% of cases that include at least some commons element, Shareable's [9] effort is a clear step towards dissipating "people's blindness to the commons." By transforming our understanding of how resources are shared and produced by communities [62], the commons paradigm is emerging as a foundation of the sharing transformation for just and sustainable cities, adding to a growing body of work [25,31,60,63,64]. However, as results showed, labor processes and relationships to nature were the two categories least impacted in *Sharing Cities* [9] cases. In other words, the two categories that have been at the core of much progressive socio–political movements of the second half of the 20th century would be the least concerned by the Sharing Cities narrative, a major blow to its promoters whose ambition is to promote "just and sustainable cities" [1].

This contradiction needs to be discussed. Of course, on a methodological level, one can argue that Shareable's account missed out on specific practices. Coworking is overlooked despite it having become mainstream and having been described as a new urban infrastructure enabling community-based collaboration and social relationships for otherwise isolated workers [65]. Likewise, the present review of cases may have been too conservative—which is difficult to estimate, given the low operationalization of arenas of impacts.

More certainly though, this contradiction may have epistemological reasons. Indeed, categories such as work/labor or nature may well have been too narrowly conceived: As an illustration, the chapter dedicated to work in *Sharing Cities'* (2018) does not include any example of care labor or domestic activities. A feminist perspective on work and the economy, however, has demonstrated that the reproduction of work and social life is made possible only through unpaid domestic work and other practices of care that are generally not seen as labor [66,67]. As for nature, ecofeminists point out that, rooted in classical and dualist ontologies, classical economic epistemologies systematically

ignore the contributions of non-humans and see in the duality of nature/culture, a main cause for human (over) exploitation of the non-human [68]. In contrast, understood in the context of a relational epistemology [33,69], commoning is a more-than-human phenomenon [70] and invites us to reconsider the basic tenets of analysis beyond the classical object/subject, natural/human divide, or even Marxist categories:

> *"The agent of change, the commoner, is no longer (and perhaps never was) a person or a category such as the working class but an assemblage. Certainly these assemblages include humans, but they also include non-humans; they may include class but also non-class alignments; they may include social movements and grassroots organizations but also governments, institutions and firms; they may include non-market mechanisms but also markets; they may include animate beings who have nothing in common except breathing and living, but also inanimate entities that share an existence on this planet."* [71] (p. 210)

This resonates with a more general call for sustainability science to adapt to the reality of the Anthropocene where the natural cannot be distinguished anymore from human influence [72,73]. With this relational framing, identifying commoning shifts the focus from a shared resource, its associated practices and impacts thought of as separate units of analysis to a commoning-community where the commons itself is the measure of success [71]. To illustrate this alternative epistemological viewpoint, it is useful to look briefly at three cases from *Sharing Cities* (2018) in a new light.

- Foodcloud in Ireland is a simple app that allows for the redistribution of surplus food to people in need [9] (p. 82). In this article, the case was reviewed and ranked as not affecting relations to nature: Food being understood as a human commodity. In contrast, a more-than-human commoning perspective makes obvious that plants play a key role in the availability of (surplus) food for people in need. It is the partnership of a commoning-community formed by people in need, local businesses, activists, plants produce, a digital app system, and a supportive legal environment that allow a commons of consumable and affordable surplus food to emerge with a strong local sustainability impact: Indeed, since its inception, Foodcloud distributed 20 million meals and diverted 9000 tons of waste from landfill [9] (p. 82).

- The Urban Agriculture Incentive Zone in San Francisco [9] (p. 92) shows how one sharing initiative may transform at the same time many established—and interdependent—dimensions of life in the city by facilitating the coexistence of functions that are usually considered as separate in the Global North: Leisure versus labor, productive vs. reproductive time and space, residential vs. agriculture, private garden vs. public space, city vs. nature, etc. Here, by accepting that commoning is not to be reduced to questions of resource and property [71], we are able to identify a new commoning-community where naturalist epistemologies could not identify a clear-cut commons.

- In New Zealand, the national parliament granted full rights of personhood to the Whanganui River, answering a long-standing revindication of the Whanganui iwi people. This opens far-reaching possibilities for the latter: To ensure the protection of the river it derives its very name from [9] (p. 201). This restores an indigenous cosmology that conceives the identity of the human community as intertwined with the non-human (the river). This example embodies the assertion that commoning is a relational process of negotiating access, use, benefit, care and responsibility; between humans, and between humans and the non-human world around them [74,75]. Importantly, from this relational worldview, the commons are not seen as objects that pre-exist their creation but rather as generated by social relations and practice [76,77].

When seen through the lens of commoning as a relational and more-than-human reality [70] the cases stated above illustrate the (re)emergence of a relational worldview "in which people, business, economy, environment and society are no longer separate worlds that meet tangentially, but are deeply interconnected and mutually interdependent" [78] (p. xii). The latter argue that a worldview

transition requires what Scharmer and Kaufer [79] alled a shift from ego to eco-consciousness [78]. To facilitate such a transition, contemplative and mindfulness practices are seen to play a catalyzing role [78,79]. Interestingly, Doran [80] argued that a surge of mindfulness practices can be observed in society and may contribute to the creation of spaces for commoning. These deserve attention for their potential contribution as and to commoning practices that bring about just and sustainable Sharing Cities; an orientation that would bridge this urban narrative with exciting recent work, stressing the potential contribution of mindfulness practices to sustainable development in relation to education [81], organizations [82] or behavior [83,84].

## 6. Conclusions

From a case review that used McLaren and Agyeman's [1] Sharing Cities approach and examined the 66 cases depicted by Shareable [9] that describe a practice (and not a policy), results showed that most practices presented as constituents of Sharing Cities are communal, although the degree of communality varies, with many cases involving monetary exchange, for example. Based on the whole sample of the cases (139), results also showed that sharing and associated policies involve equally tangible and intangible domains with two concentrations on (re)production facilities and capabilities, confirming the transformative focus of the Sharing Cities discourse rather than the transaction-centered approach characteristic of the Sharing Economy. The scaling of commoning practices is seen as happening through horizontal dynamics of replication and out-scaling, but it also requires political leadership at the level of cities. In this regard, resources depicting concrete and replicable commoning practices are seen as playing a key role in performing sharing cities.

Notably, digital platforms were found not to be central in the Sharing Cities narrative—a clear contrast to the zeitgeist of the sharing economy and smart city discourse. To this end, the alternative sub-narrative of Platform/Open Cooperativism displaces the discussion from a rampant technological solutionism to elaborating new and cooperative—commoning—models, to ensure the collective ownership of digital platforms.

While most cases reviewed do display some commons elements, only less than a third depict a clear-cut commons as understood by the Ostrom tradition and its rather naturalist epistemology. However, as discussed a more-than-human and relational understanding of (urban) commoning focuses on commoning-communities [71] rather than resources and may prove more useful in understanding the nature of the sharing transformation at the intersection of the cyber and urban spaces that are characterized by complexity, as suggested in previous work [25].The discussion of this article proposes that following Gibson-Graham et al.'s [71] epistemological approach would benefit the analysis of sharing and commoning in cities, and it would appropriately reflect the emergence of a worldview based on interdependence as a response to the challenges of the Anthropocene as identified by Ruder and Sanniti [85]. Anchoring the Sharing Cities discourse in such an understanding of commoning could answer Klein's [86] call to go beyond the articulation of a set of policy proposals and practices by exploring avenues to translate an alternative and emerging worldview based on interdependence, reciprocity, and cooperation into the urban context.

Concretely, in a context where traditional approaches to sustainability show their limits in the face of the everyday reality of the Anthropocene [73,87,88], research on ways to foster sustainability in and from cities may need to shift gear towards more radical epistemological approaches. This can take the shape of a research agenda that is informed by a relational epistemology, which seeks to identify commoning-communities in urban contexts [71], building upon the diverse economies research program achievements [89]. As an illustration of the widening of scope needed, it may be useful to document—as component of the Sharing Cities narrative—the contribution of contemplation and mindfulness practices to birthing a more-than-human worldview, possibly depicting urban commoning-communities that cater to contemplative commons [80].

**Supplementary Materials:** The supplementary materials are available online at http://www.mdpi.com/2071-1050/11/16/4358/s1.

**Author Contributions:** The author works on all parts of the research including framing of the study, collection and analysis of documents, and writing of the results.

**Funding:** We acknowledge support by the German Research Foundation (DFG) and the Open Access Publication Fund of Humboldt-Universität zu Berlin for covering the article processing charges.

**Conflicts of Interest:** The author took part in the collective authoring of the book reviewed in the present article *Sharing Cities* (2018). Otherwise, the author declared no conflict of interest.

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
