# Peer review of "Sharing Cities and Commoning: An Alternative Narrative for Just and Sustainable Cities"

_sustainability, doi:10.3390/su11164358_

Round 1
Reviewer 1 Report
· The main themes of the paper is quite interesting.
· The paper is quite well-designed and it also uses adequate relevant information and statistical data for the research and presentation of the particular subject.
· The paper is quite interesting and significantly covers the particular settings and appropriate specifications.
· The section of conclusions of this paper is quite clear enough, including the main – bullet points contributing substantially towards the current theory .
Author Response
Thank you for the review.
Reviewer 2 Report
Reviewer’s comments for Sharing Cities and Commoning: An Alternative Narrative for Just and Sustainable Cities
1. Addressing the trendy topic of sharing cities, this research paper is both well-written and engaging. Using the framework by McLaren and Agyeman (2015), the author reviewed a total of 137 secondary cases from Shareable (2018) to draw a number of interesting insights that adds in no small ways to the growing body of literature on sharing cities.
2. In Line 87, the author juxtaposed the sharing paradigm with utopian discourses as set forth in the Fourth Industrial Revolution by Rifkin (2014). A more appropriate juxtaposition would be with that typically surfaced in the smart city narrative. Smart cities around the world see digital technology as the panacea to their urban problems and that techno-utopian view still lingers even though increasingly, city government are moving away from it. Rifkin (2014), on the other hand, was actually referring to a whole array of novel technologies which may or may not be relevant to the author’s argument. As such, the author’s case would be much stronger if he contrasts the sharing paradigm to the smart city archetype.
3. In Subsection 4.1, it is interesting to note that the book failed to feature cases from Southeast Asia and China. This is quite extraordinary in that the sharing movement has gained much traction in Southeast Asia and China in recent years, primarily as a response to rapid urbanization. This being the case, the author should explain or at least account for this inconsistency.
4. In Line 629, there is a grammatical mistake. The author wrote “…scholarship that conceptualized an the role….”. The word “an” should be deleted.
5. In Line 637 to 638, the author wrote “… (growth, consumerism, profit maximization) …”. The word “and” should be add between “consumerism” and “profit maximization”.
6. From Line 654 to 660, the author made references to “distributed blockchain technology”. Often used interchangeably, blockchain and distributed ledger are actually not the same. Presumably, the author is referring only to blockchain technology in that passage. It helps to make this point clear.
7. Focusing on the concluding section, the author alluded to how the Sharing Cities narrative can help shape the dominant world view based on competition and consumerism (see Line 836 to 839). It is important to recognize that the insights draw from this study is based on a review of 69 out of 137 cases (or 50.4 per cent). It is recommended that the author acknowledges this major limitation and hence, tone down some of his/her sweeping statements/conclusions.
Author Response
Thank you for these invaluable comments.
Here are my responses to the various points:
2. Followed the recommendations and replaced the reference to Rifkin by reference to critiques of the utopian nature of the smart city approach.
3. In section 4.1 I mentionned the fact that cases are missing from China and South-East Asia in spite of the emergence of the sharing movement.
4. and 5. corrected grammar.
6. No reference is made to "distributed ledgers". Only mention of "distributed blockchain technology". Blockchain technology being a specific type of distributed ledger, I think it is adequate to add the adjective "distributed" to stress a particular quality, in particular to draw the attention of the uneducated reader.
7. Half the cases were reviewed for the sharing spectrum because the other half were not eligible as such, describing local policies. While reviewing the sharing domain as well as the existence of a commons, all cases (including local policies) were concerned. Taken this into account, the review may be more substantial than implied by the reviewer. However, I agree that my conclusions needed to be toned down, more in tune with the actual results and I conducted revisions in this direction better delineating my ideas from existing work by others.
Reviewer 3 Report
The purpose of the research is clearly defined and creates the reader's expectations on four levels: defining the studied phenomenon, the role of technology in the sharing practices, the resulting transformations and the represented goods.
The title of the article is also informative and relevant. The 89 bibliographic references are largely recent and integrate reference books for nearly 3 decades that have shown value over time.
The introduction and background of the research are consistent with the research and include both the state of knowledge in the field and the review of the specialized literature. But the question is why the author felt the need to make bibliographic references, including the conclusions.
The research methodology is an original one that combines the analysis of the specialized literature with the application of a questionnaire to limit subjectivism. However, there is a need to develop the presentation of the methodology in order to increase the credibility of the results obtained and allow other researchers to apply these methods in the event that they would propose to replicate the study.
Besides, in the presentation of the obtained results, the figures and the tables are not assumed by the author, their source not being specified.
Discussions and conclusions analyze results from multiple angles and respond in part to the object of the research. The author may insist more on the limits and potential risks of research, and make it more clear if and how it integrates into his doctoral research.
In conclusion, my recommendations are as follows:
- Detailed research methodology to allow replication of research and increase the chances of quoting the article;
- the delimitation in conclusions of ideas and value judgments belonging to the author towards the ideas of the cited authors;
- Specifying the sources of figures and tables.
Author Response
Thank you for these invaluable comments. Here are my responses.
I did not understand the following reviewer's comment: "But the question is why the author felt the need to make bibliographic references, including the conclusions."
Authorship to the tables was added.
In the conclusion section, some of the more ambitious statements were toned down. Additional references to authors were made to better delineate the contours of my own ideas and existing work from others (in particular Gibson-Graham et al. 2016).
Regarding the methodology. I applied small additions to clarify the methodology. Beyond these additions, I do not see more opportunity to describe the methodology in more details: the definition of the sharing domain was not conducted through closed questions for example. However, in order to facilitate and encourage the reproduction of results, I attach my database (a table) with the detailed scoring of each case and question. This can be made available to others on request or, possibly, as an annex by the journal.